# Multi-Head Mixture-of-Experts

**Xun Wu, Shaohan Huang**[✉]**, Wenhui Wang, Shuming Ma, Li Dong, Furu Wei**
Microsoft Research Asia
xunwu@microsoft.com, shaohanh@microsoft.com, fuwei@microsoft.com
https://aka.ms/GeneralAI

## Abstract

Sparse Mixtures of Experts (SMoE) scales model capacity without significant increases in computational costs. However, it exhibits the *low expert activation* issue, i.e., only a small subset of experts are activated for optimization, leading to suboptimal performance and limiting its effectiveness in learning a larger number of experts in complex tasks. In this paper, we propose Multi-Head Mixture-of-Experts (MH-MoE). MH-MoE split each input token into multiple sub-tokens, then these sub-tokens are assigned to and processed by a diverse set of experts in parallel, and seamlessly reintegrated into the original token form. The above operations enables MH-MoE to significantly enhance expert activation while collectively attend to information from various representation spaces within different experts to deepen context understanding. Besides, it's worth noting that our MH-MoE is straightforward to implement and decouples from other SMoE frameworks, making it easy to integrate with these frameworks for enhanced performance. Extensive experimental results across different parameter scales (300M to 7B) and three pre-training tasks—English-focused language modeling, multi-lingual language modeling and masked multi-modality modeling—along with multiple downstream validation tasks, demonstrate the effectiveness of MH-MoE.

## 1  Introduction

Large capacity models, such as Large Language Models (LLMs) [39, 28, 6, 25] and Large Multi-modal Models (LMMs) [37, 27], have demonstrated their efficacy across various domains and tasks. To further enhance performance, a reliable approach involves scaling up these models by augmenting the parameter count [13]. But for most of these densely-activated large-capacity models (referred to as Dense models), which utilize all their parameters to process all inputs, the extremely large size of these models significantly reduces inference speed, further limiting their practicality.

A promising alternative, facilitating model scalability while mitigating the burdensome computational costs, resides in Sparse Mixtures of Experts (SMoE) [31, 12, 5, 7]. In contrast to Dense model, SMoE contains parallel feed-forward neural networks (referred to as experts) within each building block, and strategically activates distinct experts for specific input tokens via a router, thereby yielding noteworthy efficiency enhancements. For instance, GShard [21] scales a Dense model from 2B to 600B parameters with lower training costs than a 100B Dense model. And recently, Mixtral $8\times7B$ [16], a SMoE model containing 8 experts is shown to outperform or matches LLaMA-2 70B [34] and GPT-3.5.

Despite its success, SMoE exhibits the *low experts activation* issue, which means that only a small subset of experts are activated during optimization and inference, e.g., **8.33%** activation ratio[1] shown in Figure 1 (a), while the majority of them are not used at all (see the dark area). As a result, SMoE

---

[1]Experts activation ratio shown in Figre 1 (a) is the ratio of each expert's selection frequency in each MoE layer to the total number of tokens, where those exceeding a threshold ($<1$) are considered activated.

fails to utilize the full expressive power of these experts, especially when the number of experts is large, which significantly limits effectiveness and scalability of SMoE.

Our aim in this paper is to achieve denser expert activation (i.e., better utilization of "dead" experts) without increase in computational cost. To achieve this, we propose Multi-Head Mixture-of-Experts (MH-MoE). The workflow of MH-MoE is illustrated in Figure 2. Inspired by the multi-head mechanism utilized in Multi-Head Self-Attention (MHSA) block, MH-MoE splits each input token into multiple sub-tokens and distribute them to different experts. After expert processing, sub-tokens are seamlessly reintegrated into the original token form, thereby achieving denser expert activation, e.g., **90.71%** activation in Figure 1 (a), while also circumventing any additional computational burden in subsequent non-parallel layers, e.g., MHSA block. Specifically, as shown in Figure 2, when provided with a single input token, MH-MoE activates four experts by splitting it into four sub-tokens, whereas SMoE only activates a single expert.

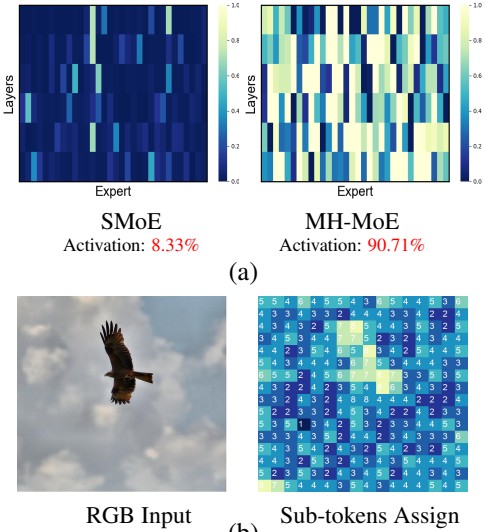

SMoE
Activation: 8.33%

MH-MoE
Activation: 90.71%

(a)

RGB Input

Sub-tokens Assign

(b)

Figure 1: (a) **Expert activation distribution** on XNLI [10] corpus, encompassing 6 parallel expert layers with 32 experts per layer. (b) **MH-MoE showcases finer-grained understanding** by distributing sub-tokens split from semantically-rich patches to more distinct experts to capture semantic information. Brighter regions indicate that sub-tokens split from this patch are distributed to a greater number of different experts.

Furthermore, we observe an interesting phenomenon: for tokens with richer semantic information, the sub-tokens split from these tokens are more likely to be allocated to distinct experts. For example, refer to the bright area in Figure 1 (b), where sub-tokens split from these patches are allocated to a greater number of different experts, facilitating the capture of semantically-rich information (e.g., the eagle in the figure). We therefore speculate that the allocation of sub-tokens to distinct experts enables MH-MoE to simultaneously focus on information from various representation spaces within different experts, ensuring a more granular understanding for subtle differences in both vision and language patterns, finally achieving better finer-grained understanding ability. See in Figure 2, sub-tokens assigned to Experts 3 and 2 capture a detailed understanding of each character's actions within an image patch, while those assigned to Experts 1 and 4 explicitly model the semantics of the false cognate 'camera'.

MH-MoE maintains following strengths: (1) **Higher experts activation & better scalability**. MH-MoE can alleviate lower expert activation problem and significantly enhance the usage of larger experts by enabling optimization of almost all of experts, allowing for more efficient scaling of model capacity. (2) **Finer-grained understanding ability**. By adaptively assigning sub-tokens to different experts based on the semantic richness of the token, MH-MoE enabling to jointly attend to information from different representation spaces at different experts, and finally achieving better finer-grained understanding ability. (3) **Seamless integration**. The implementation of MH-MoE is remarkably straightforward and decoupled from other SMoE optimization methods (e.g., GShard [21]), making it very easy to integrate them together to achieve better performance.

We evaluate the proposed MH-MoE on three model pre-training and fine-tuning setting: English-focused language modeling, multi-lingual language modeling and masked multi-modality modeling, across different parameter scales (300M to 7B). Extensive experimental among these three tasks demonstrate the effectiveness of MH-MoE.

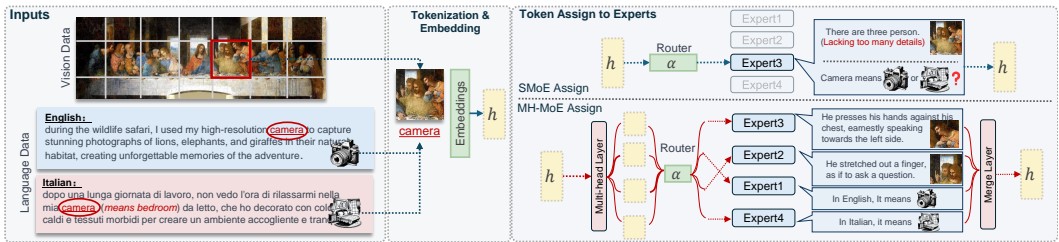

Figure 2: **Workflow of MH-MoE**. For vision data, different heads routed to different experts try to capture different aspects of details within patches and relations between patches. For language data, different heads attend to capture the varying contexts of false cognates across different languages (e.g., Italian and English) or polysemous words within a single language.

## 2 Background

Sparse Mixture-of-Experts (SMoE) [31, 12, 5, 7] enhances model capacity while maintaining a constant computational demand, thus achieving better performance than densely-activated models on various tasks [22, 19, 39, 28].

Different from densely-activated models, each MoE layer consists of $N$ independent Feed-Forward Networks (FFN) $\{f_i^{\text{FFN}}\}_{i=0}^{N}$ as the experts, along with a gating function $g(\cdot)$ to model a probability distribution indicating the weights over these experts' outputs. For the hidden representation $\mathbf{h} \in \mathbb{R}^d$ of each input token, the gating value of routing $\mathbf{h}$ to expert $f_i^{\text{FFN}}$ is denoted as:

$$g\left(f_i^{\text{FFN}}\right) = \exp\left(\mathbf{h} \cdot \mathbf{e}_i\right) / \sum_{j=0}^{N} \exp\left(\mathbf{h} \cdot \mathbf{e}_j\right), \tag{1}$$

where $\mathbf{e}_i$ denotes the trainable embedding of the $i$-th expert and $\sum_{i=0}^{N} g\left(f_i^{\text{FFN}}\right) = 1$. Then, the corresponding $k$ experts, according to the top-$k$ gated values, are activated and the output $\mathbf{O}$ of the MoE layer is $\mathbf{O} = \mathbf{h} + \sum_{i \in \Phi} g\left(f_i^{\text{FFN}}\right) \cdot f_i^{\text{FFN}}\left(\mathbf{h}\right)$, where $\Phi$ denote activated experts set and $|\Phi| = k$.

As described above, the most commonly used routing mechanism involves selecting the top-$k$ experts from $N$ experts, where $k \ll N$ [32], e.g., $k = 2$ and $N = 2048$ in GShard [21]. Such a routing mechanism allows the combination of data parallelism and expert parallelism. Some works [38, 21] suggest that larger values of $k$ often contribute to better model performance. However, with the increase in the value of $k$, training models with conventional top-$k$ routing implementation becomes much less efficient [21]. In this paper, we introduce MH-MoE, a simple but efficient manner to make denser expert activation without an increase in computational complexity.

## 3 Method

### 3.1 Multi-Head Mixture-of-Experts

Concretely, we denote a sequence of inputs tokens by $\mathbf{X} \in \mathbb{R}^{l \times d}$, where $l$ is the number of tokens and $d$ represents the length of token dimension. In MH-MoE, each parallel layer contains a set of $N$ experts, each presented as $\{f_i^{\text{FFN}} : \mathbb{R}^{\frac{d}{h}} \to \mathbb{R}^{\frac{d}{h}}\}_{i=0}^{N}$, where $h$ denotes the number of heads (i.e., the number of sub-tokens a single token is split into), which is decoupled from the head in the multi-head self-attention layer. For clarity, we describe the operation of a single MH-MoE layer here only.

The full architecture of MH-MoE can be seen in Figure 3. First, $\mathbf{X}$ is projected by a multi-head layer with parameter matrices $\mathbf{W}_{\text{head}} \in \mathbb{R}^{d \times d}$,

$$\hat{\mathbf{X}} = \mathbf{X} \cdot \mathbf{W}_{\text{head}}^{\top} \tag{2}$$

where $\hat{\mathbf{X}} \in \mathbb{R}^{l \times d}$. After that, every token in $\hat{\mathbf{X}}$ is split into $h$ sub-tokens along the token dimensions, and these sub-tokens are arranged in parallel according to the original token sequence, forming a new

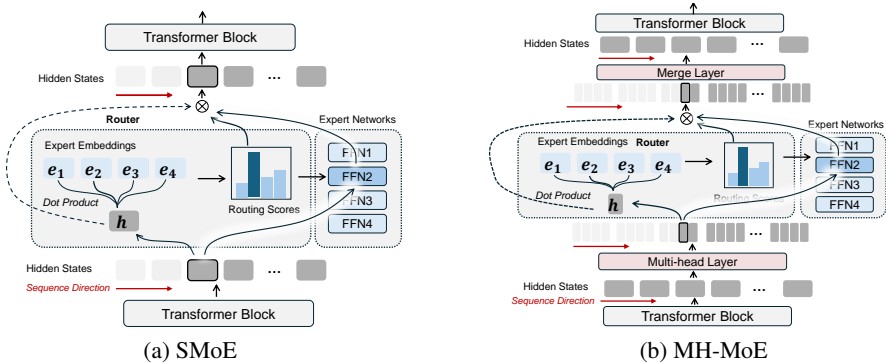

(a) SMoE
(b) MH-MoE

Figure 3: **Illustration of a typical SMoE layer and the proposed MH-MoE layer**. (a) A SMoE layer consists of a router and expert networks, where the experts are sparsely activated according to dot-product token-expert routing scores. (b) MH-MoE introduces additional two MLP layers, namely the multi-head layer and merge layer to split and merge tokens, respectively.

feature space $\ddot{\mathbf{X}} \in \mathbb{R}^{(l \times h) \times \frac{d}{h}}$ as[2]:

$$\ddot{\mathbf{X}} = F_s(\hat{\mathbf{X}}) = \left[ \overbrace{\mathbf{x}_0^0, \ldots, \mathbf{x}_{h-1}^0}^{h}, \ldots, \overbrace{\mathbf{x}_0^i, \ldots, \mathbf{x}_{h-1}^i}^{h}, \ldots, \overbrace{\mathbf{x}_0^l, \ldots, \mathbf{x}_{h-1}^l}^{h} \right]_{l \times h}, \tag{3}$$

where function $F_s$ denotes the token splitting operation: $\mathbb{R}^{l \times d} \to \mathbb{R}^{(l \times h) \times \frac{d}{h}}$, and each sub-token is presented as $\mathbf{x}_j^i \in \mathbb{R}^{\frac{d}{h}}$, meaning it is the the $j^{th}$ sub-token split from the $i^{th}$ token. Then all these sub-tokens are fed into the gating function $g(\cdot)$. The gating value of routing a certain sub-token $\mathbf{x}_j^i$ into the $p^{th}$ expert is computed as

$$g\left(f_p^{\text{FFN}}\right) = \exp\left(\mathbf{x}_j^i \cdot \mathbf{e}_p\right) / \sum_{\xi=0}^{N} \exp\left(\mathbf{x}_j^i \cdot \mathbf{e}_\xi\right), \tag{4}$$

where $\mathbf{e}_p \in \mathbb{R}^{\frac{d}{h}}$ is the learnable embedding of the $p^{th}$ expert. In this paper, we mainly focus on top-$k$ routing, *i.e.*, only the experts with the largest top-$k$ routing score is activated. $\Phi = \text{Top}_k\left(g\left(f^{\text{FFN}}\right)\right)$ denote the set of activated experts and $|\Phi| = k$. Then $\mathbf{x}_j^i$ is processed by these activated experts as following,

$$\mathbf{o}_j^i = \mathbf{x}_j^i + \sum_{p \in \Phi} g\left(f_p^{\text{FFN}}\right) \cdot f_p^{\text{FFN}}\left(\mathbf{x}_j^i\right). \tag{5}$$

After that, all obtained $\mathbf{o}_j^i$ are rearranged in the original order of sub-tokens and concatenated together as[2]:

$$\mathbf{O} = \left[ \overbrace{\mathbf{o}_0^0, \ldots \mathbf{o}_{h-1}^0}^{h}, \ldots, \overbrace{\mathbf{o}_0^i, \ldots, \mathbf{o}_{h-1}^i}^{h}, \ldots, \overbrace{\mathbf{o}_0^l, \ldots, \mathbf{o}_{h-1}^l}^{h} \right]_{l \times h}, \tag{6}$$

where $\mathbf{O} \in \mathbb{R}^{(l \times h) \times \frac{d}{h}}$. After that, $\mathbf{O}$ is transformed back the into original token form by a token merging operation $F_m: \mathbb{R}^{(l \times h) \times \frac{d}{h}} \to \mathbb{R}^{l \times d}$ and then projected by a merge layer with parameter matrices $\mathbf{W}_{\text{merge}} \in \mathbb{R}^{d \times d}$ to effective integration of multiple features $\mathbf{o}_j^i$ capturing detailed information from different expert representation spaces:

$$\check{\mathbf{X}} = F_m\left(\mathbf{O}\right)^\top \cdot \mathbf{W}_{\text{merge}}^\top. \tag{7}$$

Then we get the final output $\check{\mathbf{X}}$ of the single MH-MoE layer.

By implementing the aforementioned operations, we effectively increase the average volume of data routed to a specific expert by a factor of $h$ (as demonstrated in Eq. 3), thus achieving denser expert

---

[2]Sub-tokens within regions marked with the same color are split from the same token.

activation. Besides, the shapes of the input and output in the MH-MoE layer remain unchanged, thus no additional computational cost is introduced in the subsequent block. Specifically, we introduce a hyperparameter $\beta$ to scale the inner dimensions of each expert, aiming to balance the parameters introduced by the multi-head layer and merge layer, aligning the model's parameters and computational complexity with the original SMoE. Furthermore, the allocation of sub-tokens to distinct experts within MH-MoE enables us to collectively capture semantic information from diverse feature spaces across these experts, thereby enhancing the model's ability to achieve a finer-grained understanding.

As the Pytorch-like style pseudocode of MH-MoE shown in Appendix D, MH-MoE is characterized by its overall simplicity of implementation and decoupled from other SMoE optimization strategies [21, 5], making it easy to integrate with other optimized SMoE frameworks to enhance performance.

### 3.2 Training Objectives

**Load balancing loss.** To mitigate the expert load imbalance issue, given the sub-token set $\ddot{\mathbf{X}}$ (depicted in Eq. 3) and the frequency $t_p$ of how many sub-tokens are routed to the $p^{th}$ expert, we follow existing works [21, 13] to compute the load balancing loss $\mathcal{L}_{\text{balance}}$ via:

$$\mathcal{L}_{\text{balance}} = \frac{N}{|\ddot{\mathbf{X}}|} \sum_{p=1}^{N} \sum_{\mathbf{x}_j^i \in \ddot{\mathbf{X}}} t_p \cdot g\left(f_p^{\text{FFN}}\right), \tag{8}$$

where $N$ denotes the number of experts, $|\ddot{\mathbf{X}}|$ is the number of sub-tokens contained in $\ddot{\mathbf{X}}$. $g\left(f_p^{\text{FFN}}\right)$ denotes the gating value of routing a certain sub-token $\mathbf{x}_j^i$ into the $p^{th}$ expert (see in Eq. 4).

**Task specific loss.** The term $\mathcal{L}_{\text{task}}$ is dependent on the particular task that MH-MoE is designed to learn. For instance, during pre-training in the English-focused Language Modeling task, we utilize the language modeling loss [29], whereas the model predicts the next word in a sequence.

So, the overall training objective of MH-MoE is to minimize:

$$\mathcal{L} = \mathcal{L}_{\text{task}} + \alpha \mathcal{L}_{\text{balance}}, \tag{9}$$

where $\alpha$ is a coefficient for load balancing.

## 4 Experiments

### 4.1 Experimental Setup

**Compared Baselines.** We include two baseline models for comparison: (1) **Dense**, which represents a Transformer decoder without the incorporation of sparsely-activated parallel modules (i.e., SMoE layer). (2) **X-MoE**, which is our implementation based on the SMoE proposed by [5]. We build MH-MoE upon X-MoE and use identical settings. Note that the all models are pre-trained using the same training data and loss (Eq. 9) as MH-MoE, and we ensure that the parameter count of MH-MoE remains consistent with or lower than that of X-MoE, ensuring a fair and equitable comparison. A detailed comparison about parameter and computational complexity can be found in Table 11.

**Pre-training Data.** We detail the pre-training data of MH-MoE in three areas: (1) English-focused experiments use the RedPajama dataset [8], which is an open-source pre-training dataset. (2) multi-lingual tasks follow XLM [20] and use the multilingual Wikipedia as training data. (3) multimodal tasks use a masked multi-modality modeling task with a large dataset of images, documents, and image-text pairs. Further details are available in the Appendix A.

**Model Architecture and Hyperparameters.** For all experiments, we use the X-MoE [5] as our backbone architecture to build our MH-MoE, which has shown better performance than prior SMoE models such as Switch Transformers [13] on cross-lingual understanding benchmarks. For English-focused Language Modeling and Multi-lingual Language Modeling, we construct Dense, X-MoE and MH-MoE using the Transformer [36] decoder (L = 12, H = 768, A = 12) with the GPT-4[3] vocabulary

---

[3]`https://github.com/openai/tiktoken`

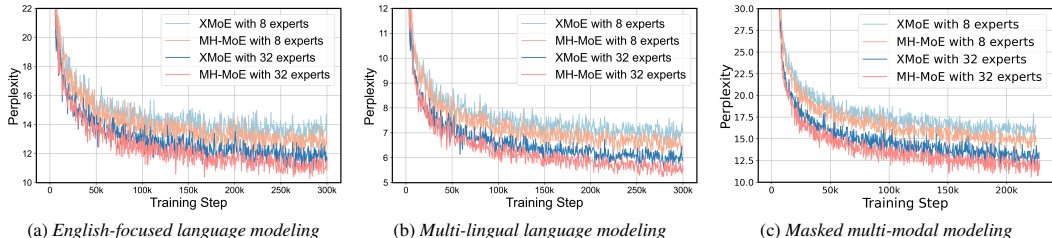

|  | (a) *English-focused language modeling* | (b) *Multi-lingual language modeling* | (c) *Masked multi-modal modeling* |

Figure 4: **Perplexity on validation dataset during the training phase** reported for Dense, X-MoE and MH-MoE across three pre-training tasks.

Table 2: Accuracy / accuracy-normalization scores for language understanding tasks using the LLM Evaluation Harness [14]. $N$ denotes the number of experts.

| Model | ARC-Challenge | ARC-Easy | RTE | BookQA | Winogrande | PiQA | BoolQ | HellaSwag | TruthfulQA (mc1/mc2) | Avg |
|---|---|---|---|---|---|---|---|---|---|---|
| Dense | 18.1/23.3 | 44.9/39.7 | 51.5 | 17.1/29.0 | 48.2 | 66.6 | 55.0 | 29.7/34.1 | 24.1/39.3 | 37.2 |
| X-MoE ($N = 8$) | 19.0/24.7 | 48.3/42.0 | 52.7 | 17.4/29.8 | 50.3 | 67.9 | 58.4 | 31.4/35.7 | 24.3/40.2 | 38.7 |
| MH-MoE ($N = 8$) | **19.6/25.2** | **50.2/42.2** | **53.0** | **18.2/30.3** | **51.1** | **68.7** | **59.6** | **33.2/40.3** | **24.7/40.9** | **39.8** |
| X-MoE ($N = 32$) | 19.4/24.8 | 50.4/42.5 | 52.7 | 17.8/30.0 | 51.3 | 68.8 | 52.8 | 33.4/40.1 | 24.3/39.1 | 39.1 |
| MH-MoE ($N = 32$) | **21.4/26.8** | **50.6/44.8** | **53.4** | **18.8/31.6** | **53.8** | **69.3** | **56.6** | **35.0/42.1** | **24.8/39.5** | **40.6** |

as the backbone architecture. The pre-training procedure takes 14 days on 2 NVIDIA DGX-2 Stations. For Masked Multi-modal Modeling, we build Dense, X-MoE and MH-MoE following the same Transformer encoder architecture as BEiT v3 [37]. The pre-training procedure takes 4 days on 2 NVIDIA DGX-2 Stations. For all three pre-training tasks, we set the head number $h = 4$. More details about architecture and training hyperparameters can be found in Appendix B and C.

## 4.2 Perplexity Evaluation

We examined the validation perplexity curves for all pre-trained models and pre-training tasks under two expert settings (8 experts and 32 experts). The perplexity trends are depicted in Figure 4, with the final perplexity values listed in Table 1. We can observe that as training progresses: 1) the perplexity of our MH-MoE remained lower in comparison to the compared baselines, indicating more effective learning; 2) MH-MoE achieved the lowest perplexity across three distinct experimental setups; 3) an increase in the number of experts led to a corresponding decrease in the perplexity of MH-MoE, suggesting that the model benefits from enhanced representation learning capabilities as more experts are incorporated. These results collectively demonstrate the superiority of MH-MoE in terms of learning efficiency and language representation across multiple pre-training paradigms.

Table 1: Results of upstream perplexity evaluation. We report the validation perplexity cross two setting: 8 experts and 32 experts.

| Model | Perplexity ↓ | |
|---|---|---|
| | 8 Experts | 32 Experts |
| *English-focused language modeling* | | |
| Dense (without Experts) | 16.23 | 16.23 |
| X-MoE | 14.82 | 11.96 |
| MH-MoE (Ours) | **12.72** | **10.28** |
| *Multi-lingual language modeling* | | |
| Dense (without Experts) | 8.56 | 8.56 |
| X-MoE | 7.19 | 6.02 |
| MH-MoE (Ours) | **6.26** | **5.09** |
| *Masked multi-modal modeling* | | |
| Dense (without Experts) | 17.95 | 17.95 |
| X-MoE | 16.34 | 12.68 |
| MH-MoE (Ours) | **14.73** | **10.87** |

## 4.3 Downstream Evaluation

**English-focused Language Modeling.** We evaluate our models on a total of 9 different zero-shot benchmarks to assess their abilities across various natural language tasks like common sense reasoning, general language understanding and knowledge understanding using the LLM Evaluation Harness [14]. As shown in Table 2, comparing X-MoE with the Dense model, X-MoE show notable improvement, indicating that SMoE models (e.g., X-MoE) benefit from the large model capacity. Overall, for all benchmarks, our MH-MoE obtains the best performance, achieving an average performance gain of 1.1 for 8 experts setting and 1.5 for 32 experts setting compared to X-MoE, demonstrating the effectiveness of our proposed MH-MoE on modeling English-focused language.

**Multi-lingual Language Modeling.** We evaluate our multi-lingual language models on the cross-lingual natural language inference (XNLI) corpus [10], which is the extension of the multi-genre NLI

Table 3: Accuracy / accuracy-normalization scores on multilingual understanding tasks using the LLM Evaluation Harness [14]. $N$ denotes the number of experts.

| Model | bg | de | el | en | es | fr | hi | ru | sw | th | tr | ur | vi | zh | Avg |
|---|---|---|---|---|---|---|---|---|---|---|---|---|---|---|---|
| Dense | 33.1 | 33.3 | 33.0 | 35.1 | 32.8 | 34.4 | 33.6 | 34.2 | 33.3 | 33.1 | 33.3 | 33.9 | 33.5 | 32.9 | 33.5 |
| X-MoE ($N = 8$) | 33.9 | **33.4** | 33.4 | 37.3 | 33.3 | 35.9 | 34.5 | 35.0 | 33.5 | 33.6 | 33.4 | 34.2 | 33.3 | 33.2 | 34.1 |
| MH-MoE ($N = 8$) | **34.4** | 33.2 | **33.9** | **40.1** | **34.0** | **36.4** | **34.6** | **35.2** | **33.8** | **34.4** | **33.3** | **34.7** | **34.6** | **33.5** | **34.7** |
| X-MoE ($N = 32$) | 34.5 | 34.5 | 33.4 | 39.6 | 33.1 | 35.3 | 34.1 | 35.4 | 33.6 | 34.7 | 33.7 | 33.6 | 34.5 | 33.3 | 34.5 |
| MH-MoE ($N = 32$) | **35.8** | **35.6** | **34.1** | **40.7** | **33.9** | **36.7** | **34.4** | **36.3** | **34.3** | **36.0** | **34.1** | **34.3** | **35.2** | **33.6** | **35.5** |

Table 4: Results of visual question answering, visual reasoning, and image captioning tasks.

| Model | VQAv2 | | NLVR2 | | COCO Caption | | | |
|---|---|---|---|---|---|---|---|---|
| | test-dev | test-std | dev | test-P | B@4 | M | C | S |
| Dense | 65.9 | 66.0 | 73.8 | 74.2 | 35.9 | 29.3 | 120.5 | 19.6 |
| X-MoE | 68.4 | 69.7 | 75.5 | 76.1 | 38.1 | 30.2 | 122.9 | 21.3 |
| MH-MoE | **70.1** | **71.4** | **77.0** | **77.8** | **39.7** | **33.1** | **124.1** | **23.0** |

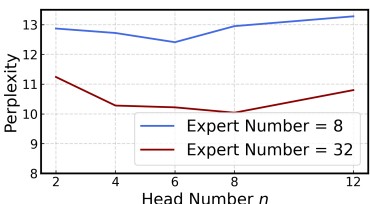

Figure 5: Comparison results for different head number $h$.

(MultiNLI) corpus to 14 languages. We follow the LLM Evaluation Harness pipeline and use the zero-shot setting to evaluate the multi-lingual ability. Table 3 presents the zero-shot evaluation results on XNLI task. Similarly, X-MoE benefit from the large model capacity and show notable improvement compared with Dense model. Overall, MH-MoE obtains the best performance, surpassing X-MoE by an average performance gain of 0.6 for 8 experts setting and 0.8 for 32 experts setting. Comparing MH-MoE with the X-MoE, it shows that MH-MoE models provide consistent gains on downstream tasks, demonstrating the effectiveness of our proposed MH-MoE on modeling cross-lingual natural language.

**Masked Multi-modal Modeling.** We evaluate on the widely used vision-language understanding and generation benchmarks, including visual question answering [15], visual reasoning [33] and image captioning [23]. We report *vqa-score* on VQAv2, accuracy for NLVR2. For COCO image captioning, we report BLEU@4 (B@4), METEOR (M), CIDEr (C), and SPICE (S). Table 4 presents the evaluation results. For VQA task, MH-MoE outperforms both Dense and X-MoE by a large margin, e.g., 4.24 and 1.69 points gain on test-dev split, respectively. For visual reasoning task, MH-MoE beats all these two baselines on both dev (1.5 points gain than X-MoE) and test-P split (1.7 points gain than X-MoE). For image captioning task, MH-MoE surpasses X-MoE by 4.2%, 10.2%, 9.4% in terms of B@4, M and S, respectively. Above results show that X-MoE exhibits enhanced visual information comprehension, which also validates the effectiveness of our proposed MH-MoE in capturing diverse semantic information within text-image pair data.

## 4.4 Ablation Studies

This section presents experimental analysis to demonstrate the functionality of MH-MoE. In all comparative experiments, *we ensure parameter equality across models by adjusting the internal dimensions of the experts*.

**Number of Heads $h$.** We conduct experiments by adjusting the number of heads ($h = 2$, 4, 6, 8, and 12) in MH-MoE. As shown in Figure 5, we find that across all settings of $h$, our model consistently outperforms the X-MoE. Besides, as the value of $h$ increases, we observe an initial improvement followed by a decline in our model's performance. This leads us to hypothesize that as $h$ initially increases, the enhancement in performance benefits from MH-MoE by activating a greater number of experts, thereby enhancing the model's effectiveness and capturing a wider range of fine-grained token information. However, as $h$ becomes too large, the excessive subdivision of tokens may impair their original semantic content, leading to decreased model performance.

**Does the Token-Splitting-Merging Really Matters?** The key motivation of MH-MoE is to split each token into several sub-tokens and then merge the sub-tokens after processing by experts. We use two MLP layers for the splitting and merging processes. We conduct a detailed analysis to determine whether the performance improvement is due to the Token-Split-Merging (TSM) operation or the additional MLP layers.

Table 5: Ablation studies of MH-MoE components: MLP layers and the Token-Splitting-Merging (TSM, Eq. 3 and Eq. 6) operation.

| Model | MLP | TSM | Perplexity |
|---|---|---|---|
| Dense | ✗ | ✗ | 16.23 |
| Dense$_{w/\ \text{MLP}}$ | ✓ | ✗ | 16.40 |
| X-MoE | ✗ | ✗ | 14.82 |
| X-MoE$_{w/\ \text{MLP}}$ | ✓ | ✗ | 14.77 |
| MH-MoE$_{w/o\ \text{TSM}}$ | ✓ | ✗ | 14.77 |
| MH-MoE$_{w/o\ \text{MLP}}$ | ✗ | ✓ | 13.97 |
| MH-MoE | ✓ | ✓ | **12.72** |

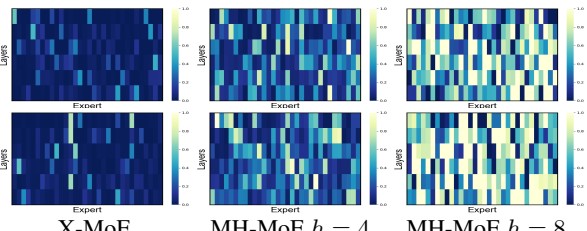

X-MoE    MH-MoE $h = 4$    MH-MoE $h = 8$

Figure 6: **Distribution of expert activation in X-MoE and MH-MoE** on both *Harness* and *XNLI* corpus, where $h$ denotes the number of heads.

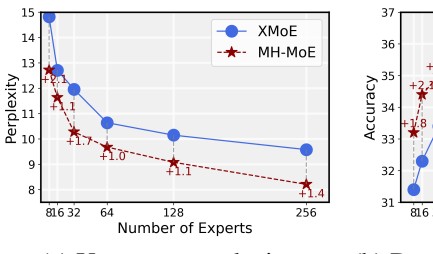 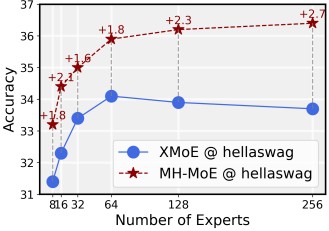 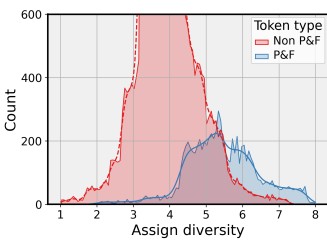

(a) Upstream perplexity    (b) Downstream accuracy scores    (c) Sub-tokens assign

Figure 7: (a) Upstream training perplexity ($\downarrow$) when scaling the number of experts for both X-MoE and MH-MoE. (b) Accuracy scores on the hellaswag task when scaling the number of experts for both X-MoE and MH-MoE. (c) Comparison for sub-tokens assign diversity (the number of different experts they are routed to) for P&F and Non P&F tokens.

The results are presented in Table 5. A comparative analysis between Dense v.s. Dense$_{w/\ \text{MLP}}$, as well as X-MoE v.s. X-MoE$_{w/\ \text{MLP}}$, reveals that introduction of the MLP layer does not enhance the model's performance. Similarly, when comparing MH-MoE with MH-MoE$_{w/o\ \text{TSM}}$, it becomes evident that the inclusion of only the MLP, in the absence of the TSM, also does not yield any improvement in the model's effectiveness. The parameter quantities of the models being compared pairwise are equal.

An intriguing observation is made when comparing MH-MoE with MH-MoE$_{w/o\ \text{MLP}}$. Introducing TSM alone, without MLP, results in a slight increase in model performance. In contrast, a significant enhancement in model performance is only achieved when both MLP and TSM are incorporated simultaneously. We hypothesize that introduction of TSM, without the integration of MLP, activates more experts, but the segmentation and merging of the model appears overly straightforward and abrupt in its execution. This limitation hinders the model's ability to meaningfully segment tokens into sub-tokens and effectively merge the diverse information gathered from different expert spaces.

We also conducted additional ablation studies to examine the **impact of varying the number of MLP layers**, **different splitting-merging methods** and **different balance loss**. The details of these experiments are provided in Appendix E.

## 5 Analysis

**Experts Activation.** We visualize the activation of each expert varies across parallel expert layers for X-MoE and MH-MoE at Figure 6. It can be observed that: 1) X-MoE demonstrate a more skewed distribution, wherein a significant portion of experts remain inactivated all the time. 2) Our MH-MoE achieves a denser expert activation compared to X-MoE, effectively mitigating the issue of low expert utilization. 3) As the number of heads $h$ increases, the expert activation frequency in MH-MoE also rises.

**Scalability.** We explore the scalability for both X-MoE and MH-MoE by scaling up the number of experts from 8 to 256 (about 7B parameters). For upstream performance, as shown in Figure 7 (a), with the increase of experts, our MH-MoE could bring more gains. It is because MH-MoE could mitigate the low expert activation problem effectively. With this ability, the superiority of the large-scale SMoE model will be better exerted, thereby achieving the improvement of the upper bound

Table 6: Performance across multiple pure vision tasks: classification (CLS) on ImageNet-1k, object detection (OD) and instance segmentation (IS) on COCO. The number of expert is set to 8.

| Model | CLS | OD | | | IS |
|---|---|---|---|---|---|
| | ACC | AP | $AP_{50}$ | $AP_{75}$ | $AP_{mask}$ |
| Dense | 70.73 | 39.81 | 58.97 | 44.46 | 36.42 |
| SMoE | 75.66 | 42.23 | 60.30 | 44.58 | 37.50 |
| MH-MoE | **77.34** | **44.45** | **63.18** | **45.85** | **38.24** |

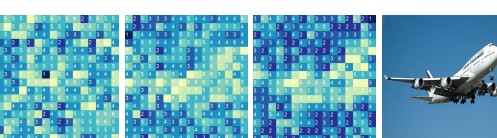

Table 7: **Assign diversity of sub-tokens split from different patches** in vision data with respect to training steps (100k → 200k → 250k steps). Brighter regions indicate sub-tokens split from this patch are distributed to a greater number of diverse experts.

of SMoE with more experts. Detailed validation perplexity curves for these scaling up experiments can be found in Figure 10 at Appendix F. For downstream performance shown in Figure 7 (b), for X-MoE, expert number = 64 is the upper bound, meaning that continuing to increase the number of experts will not bring performance gain. Our MH-MoE not only has a performance advantage over the X-MoE with the same number of experts, but also improves the upper bound from 64 to 256, which demonstrates the effectiveness of the scalability of our MH-MoE on downstream tasks.

**Experts Assign within Token.** We delve into a more granular analysis to validate how the multi-head mechanism aids MH-MoE in capturing diverse and intricate semantic information that is often challenging to comprehend, e.g., polysemous and false cognates words (denoted as PF tokens) in languages, and semantically-rich areas in images.

For languages data, we compute and compare the divergence levels (i.e., the number of different experts these sub-tokens are routed to) of sub-tokens split from PF tokens and Non-PF tokens. The results, presented in Figure 7 (c), clearly demonstrate that the distribution of divergence for PF tokens is significantly skewed towards the right when compared to that of Non-PF tokens. This indicates that during the MH-MoE's inference process, PF tokens route their sub-tokens to a greater number of different experts, thereby capturing diverse semantic information in contrast to Non-PF tokens for a better polysemous and false cognates word modeling. Note that we utilized the GPT-4 API [25] to extract polysemous words and false cognates from the XNLI [10] corpus, and the corresponding prompt can be found in Table 12.

For image data, we analyzed how the divergence levels of different patches evolve during the training process, as illustrated in Figure 7. Notably, as training progresses, divergence levels gradually increase in high-frequency texture regions (or regions with rich semantics), while they decrease in low-frequency texture regions. This suggests that MH-MoE tends to route tokens from complex texture areas to a wider variety of experts, thereby enhancing the finer-grained understanding of semantics in those regions. For more visualization examples, please refer to Figure 11 in Appendix G.

**Experiments on Pure Vision Tasks.** We conduct small-scale experiments on pure vision tasks to further validate the effectiveness of MH-MoE. We utilize ViT-B(ase) [11] as the Dense model and AdaMV-MoE [4] as the SMoE baseline. AdaMV-MoE is a multi-task vision SMoE model demonstrating a superior performance across various vision tasks. We build our MH-MoE upon AdaMV-MoE. The comparison results are shown in Table 6, which demonstrates that MH-MoE exhibits corresponding effectiveness and versatility in pure vision tasks.

**Model Collapses.** Concerns may arise that MH-MoE could revert to the original SMoE approach, routing all sub-tokens from the same token to a single expert. In MH-MoE's training framework, the load balancing loss $\mathcal{L}_{balance}$ (Eq 8) serves as a weak constraint, treating all sub-tokens as independent entities and uniformly distributing them among experts, promoting a balanced allocation. Our observations indicate that most sub-tokens from the same token are distributed among 3-5 different experts (see Figures 7 and Figures 7 (c)).

## 6 Conclusion

In this paper, we study how we can to achieve a denser experts activation without introducing additional cost, while improving the fine-grained understanding ability. With the proposed Multi-Head Mixture-of-Experts, we can easily implement the aforementioned functionality. Furthermore, the simplicity of MH-MoE allows it to integrate with other SMoE frameworks to enhance performance easily. Extensive empirical results across three tasks demonstrate the effectiveness of MH-MoE.

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

# A    Pre-training Data of Masked multi-modal modeling task

Table 8 presents of pre-training data in Masked multi-modal modeling task. For multi-modal data, there are about 15M images and 21M image-text pairs collected from five public datasets: Conceptual 12M (CC12M) [3], Conceptual Captions (CC3M) [30], SBU Captions (SBU) [26], COCO [23] and Visual Genome (VG) [17]. For monomodal data, we use 14M images from ImageNet-21K and 160GB text corpora [1] from English Wikipedia, BookCorpus [40], OpenWebText[4], CC-News [24], and Stories [35].

Table 8: Pretraining data of Masked multi-modal modeling task. All the data are academically accessible.

| Data | Source | Size |
|------|--------|------|
| Image-Text Pair | CC12M, CC3M, SBU, COCO, VG | 21M pairs |
| Image | ImageNet-21K | 14M images |
| Text | English Wikipedia, BookCorpus, OpenWebText, CC-News, Stories | 160GB documents |

# B    Model Hyperparameters of Language modeling tasks

Table 9 presents the model hyperparameters of X-MoE and MH-MoE for both English-focused language modeling and Multi-lingual language modeling tasks. The gating temperature $\tau_0$ is initialized as $0.3$ and $0.07$ for the softmax gating and sigmoid gating, respectively. We use the same vocabulary as XLM-R [9] with 250K subwords tokenized by SentencePiece [18].

Table 9: Model hyperparameters of Dense, X-MoE and MH-MoE. The SMoE frequency refers to how many experts each token will be assigned to, i.e., the value of k in the Top- expert selection.

| Hyperparameters | Dense | X-MoE | MH-MoE |
|-----------------|-------|-------|--------|
| FFNs within layer | 2 | 2 | 2 |
| Expert embedding dimension | - | 16 | $16/h$ |
| Initialized gating temperature $\tau_0$ | - | 0.3 / 0.07 | 0.3 / 0.07 |
| Transformer blocks | 12 | 12 | 12 |
| Hidden size | 768 | 768 | 768 |
| FFN inner hidden size | 3,072 | 3,072 | $3,072 \times \beta$ |
| Attention heads | 12 | 12 | 12 |
| SMoE frequency | - | 2 | 2 |

---

[4]`http://skylion007.github.io/OpenWebTextCorpus`

# C   Hyperparameters for Pre-training

Table 10 presents the hyperparameters for pre-training across three tasks: Language modeling tasks (English-focused language modeling and Multi-lingual language modeling tasks) and Masked multi-modal modeling task.

Table 10: Pre-training hyperparameters for Language modeling tasks (English-focused language modeling and Multi-lingual language modeling tasks) and Masked multi-modal modeling task tasks.

| Hyperparameters | Language modeling tasks | Multi-modality modeling task |
|---|---|---|
| Batch size | 256 | 512 |
| Optimizer | Adam | AdamW |
| Batch tokens per task | 1M | - |
| Adam $\epsilon$ | 1e-6 | 1e-6 |
| Adam $\beta$ | (0.9, 0.98) | (0.9, 0.98) |
| Maximum learning rate | 5e-4 | 2.8e-3 |
| Learning rate schedule | Linear decay | Cosine decay |
| Warmup steps | 10,000 | 10,000 |
| Weight decay | 0.01 | 0.05 |
| Transformer dropout | 0.1 | 0.1 |
| Dropout | 0 | 0 |
| Load balancing coefficient | 1e-2 | 1e-2 |

Table 11: Parameter count setting of X-MoE and MH-MoE in our experiments for English-focused language modeling, Multi-lingual language modeling and Masked multi-modality modeling tasks. "non-expert param" refers to the parameters that are not part of the expert networks, such as the attention layer, router, etc., while "expert params" represents the total number of parameters in the parallel expert networks. For Dense models, since there are no expert network layers, we only list the total number of parameters. All models under the same task utilize the same architecture and hyperparameters, following identical training settings and steps.

| Expert Setting | Dense | X-MoE | | | MH-MoE | | |
|---|---|---|---|---|---|---|---|
| | Sum | non-expert params | expert params | Sum | non-expert params | expert params | Sum |
| *English-focused language modeling* | | | | | | | |
| 0 expert | 162M | - | - | - | - | - | - |
| 8 experts | - | 134M | 227M | 361M | 141M | 213M | 354M |
| 16 experts | - | 134M | 454M | 588M | 141M | 430M | 571M |
| 32 experts | - | 134M | 908M | 1,042M | 141M | 898M | 1,039M |
| 64 experts | - | 134M | 1,815M | 1,949M | 141M | 1,797M | 1,938M |
| 128 experts | - | 134M | 3,631M | 3,765M | 141M | 3,624M | 3,765M |
| 256 experts | - | 134M | 7,263M | 7,397M | 141M | 7,230M | 7,371M |
| *Multi-lingual language modeling* | | | | | | | |
| 0 expert | 162M | - | - | - | - | - | - |
| 8 experts | - | 134M | 227M | 361M | 141M | 213M | 354M |
| 32 experts | - | 134M | 908M | 1,042M | 141M | 898M | 1,039M |
| *Masked multi-modality modeling* | | | | | | | |
| 0 expert | 277M | - | - | - | - | - | - |
| 8 experts | - | 191M | 323M | 514M | 195M | 310M | 505M |
| 32 experts | - | 191M | 1,037M | 1,228M | 195M | 1,014M | 1,209M |

# D  PyTorch-style Code

We also provide the PyTorch-style code in Algorithm 1 to explain our MH-MoE, which including two main aspects: 1) `Stage 1`. The creation and initialization of multi-head layer and merge layer. 2) `Stage 2`. The segmentation of tokens, followed by processing through an expert network, and ultimately merging.

---

**Algorithm 1** The Overall Procedures of MH-MoE in a PyTorch-like style.

---

**Input:** A MH-MoE model with L parallel SMoE layers M, the number of the experts $k$.

```python
# Stage 1: Initial parameter of multi-head layer & merge layer

for i in range(1, L):
    M[i].multi_head_layer = nn.Linear(hidden_dim, hidden_dim)
    M[i].merge_layer = nn.Linear(hidden_dim, hidden_dim)

     # Initialization
    nn.init.xavier_uniform_(M[i].multi_head_layer.weight, gain=1 / math.sqrt(2))
    nn.init.xavier_uniform_(M[i].merge_layer.weight)
    nn.init.constant_(M[i].merge_layer.bias, 0.0)

# Stage 2: The segmentation and merge of tokens for the i-th MH-MoE layer

def MHMoE_Layer(x):
    '''
    Input:
        x : Tensor shape: (batch_size, Length, hidden_dim)
        mask : Tensor shape: (batch_size, Length)

    Output:
        o : Tensor shape: (batch_size, Length, hidden_dim)

    heads: head number of multi_head layer
    '''

    # Processed by multi-head layer
    x = M[i].multi_head_layer(x)

    # Split token & rearrange sub-tokens in parallel
    x = x.reshape(batch_size * Length * heads, hidden_dim // heads).contiguous()
    mask = mask.reshape(-1, 1).repeat(1, heads).reshape(batch_size * Length * heads)

    # Standrad SMoE routing block
    x, mask = router(x, mask)

    # Merge back to the original token form
    x = x.reshape(batch_size * Length, heads, dim // heads).reshape(batch_size *
        Length, dim).contiguous()
    o = M[i].merge_layer(x)

    return o
```

---

Table 12: Prompt template for identifying polysemous and false cognates in different languages.

Your role is to identify polysemous and false cognates in different languages from the given textual input (**### Input**). Note that "Polysemous" refers to a word having two or more completely different meanings (for example, "grouse" has meanings related to complaining and also refers to a type of bird), while "false cognates in different languages" refers to words in different languages that have similar forms but carry different meanings (for example, in English and Italian, "camera" looks similar but represents different semantic concepts).

**### Input**
Text: {Prompt}

Note: Please provide your identify results in the following format:

**### Output for Word 1**
Word: [Make sure that there is only a word here.]
Rationale: [Rationale for why this word is polysemous or false cognates, think step by step]

**### Output for Word 2**
Word: [Make sure that there is only a word here.]
Rationale: [Rationale for why this word is polysemous or false cognates, think step by step]
----

Table 13: Comparison results for different numbers of MLP layers $n$. The results are averaged over five runs.

| $n$ | Upstream | Downstream | | |
|---|---|---|---|---|
| | Perplexity | RTE | PIQA | Winogrande |
| 0 | 13.97 | 52.9 | 68.2 | 51.7 |
| 1 | 12.72 | 53.4 | **69.3** | **53.8** |
| 2 | **12.66** | **54.0** | 68.8 | 53.3 |
| 3 | 12.87 | 53.1 | 68.8 | 52.7 |

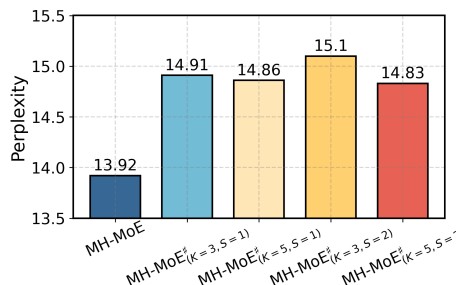

Figure 8: Comparison results for different splitting method. K denotes the size of kernel while S denotes the size of stride in Conv1D.

## E  Ablation Studies

**Number of MLP layers.** We explore the impact of varying the number of layers ($n = 0, 1, 2, 3$) in MLP on MH-MoE performance. For configurations exceeding a single layer, ReLU activation functions were incorporated between MLP layers to ensure the non-linearity of transformations. The parameter quantities of the models being compared are equal. Upon analyzing the results in Table 13, we observe that increasing the number of MLP layers beyond one had a negligible impact on the model's performance. This indicates that a single-layer MLP is sufficient for accomplishing token segmentation and fusion.

**What Makes a Good Splitting-Merging Method?** We designed ablation experiments to investigate whether the tokenization method affects the MH-MoE's performance. We replace the multi-head layer and merge layer (both are FC layers) with Conv1d layers (denoted as MH-MoE$^\sharp$).

We conducted experiments in two settings (English-focused language modeling and Multi-lingual language modeling tasks) and explored different levels of randomness by using different kernel sizes and strides for the Conv1d layers. We ensure that the parameters are kept as consistent as possible by adjusting the number of Conv1d layers and the dimensions of experts. Through experimental results presented at Figure 8, we observe that replacing splitting method with Conv1d (denoted as MH-MoE$^\sharp$) resulted in significantly inferior performance compared to the original MH-MoE. This underscores the importance of tokenization methods, indicating that tokenization methods such as Conv1d, which disrupt the original input sequence features, are not suitable for this context.

**Impact of different balance loss** We conducted experiments with three groups to assess the impact of different constraints on sub-token allocation: (1) MH-MoE$^\P$: MH-MoE trained without the $\mathcal{L}_{\text{balance}}$, (2) MH-MoE$^\dagger$: MH-MoE trained with the $\mathcal{L}_{\text{balance}}$, and (3) MH-MoE$^\ddagger$: MH-MoE trained with both $\mathcal{L}_{\text{balance}}$ and diversity loss $\mathcal{L}_{\text{div}}$. $\mathcal{L}_{\text{div}}$ is designed to enhance sub-token allocation diversity and defined as: given a set of sub-tokens $\Upsilon$ split from the same token, with $P_i$ representing the gating probability distribution of the $i^{th}$ sub-token, $\mathcal{L}_{\text{div}} = -\sum_{i=1}^{|\Upsilon|} \sum_{j=i+1}^{|\Upsilon|} \mathcal{KL}(P_i||P_j)$, where

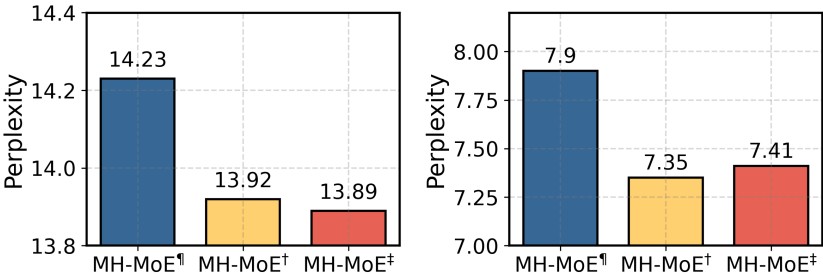

Figure 9: Experimental results for varying level constraints on sub-token allocation across two tasks: *English-focused language modeling* (left) and *Multi-lingual language modeling* (right).

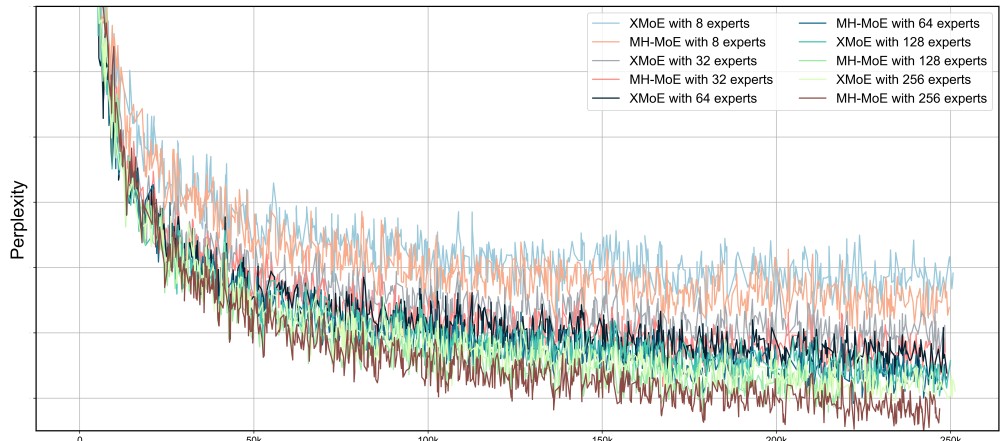

Figure 10: Validation perplexity reported for both X-MoE and MH-MoE.

$\mathcal{KL}(\cdot)$ indicate Kullback–Leibler divergence. This constraint increases the inconsistency in gating distributions among sub-tokens in $\Upsilon$, enhancing expert allocation diversity.

The experimental results (perplexity) are summarized in the Figure 9. Key findings include: (1) Without any constraints (MH-MoE[¶]), the model's performance degrades, likely due to severe imbalance, causing some experts to receive insufficient data, resulting in suboptimal performance. (2) The addition of $\mathcal{L}_{\text{div}}$ did not significantly improve model performance compared to using only $\mathcal{L}$balance. We hypothesize that $\mathcal{L}_{\text{balance}}$ alone provides sufficient diversity in sub-token allocation.

## F Visualization of training perplexity

We provide the training perplexity curve for model training in the experimental setting of increasing the number of experts (from 8 to 256) in Figure 10.

## G Visualization

We provide more visualization of variation in assign diversity for sub-tokens split from different patches in vision data at Figure 11.

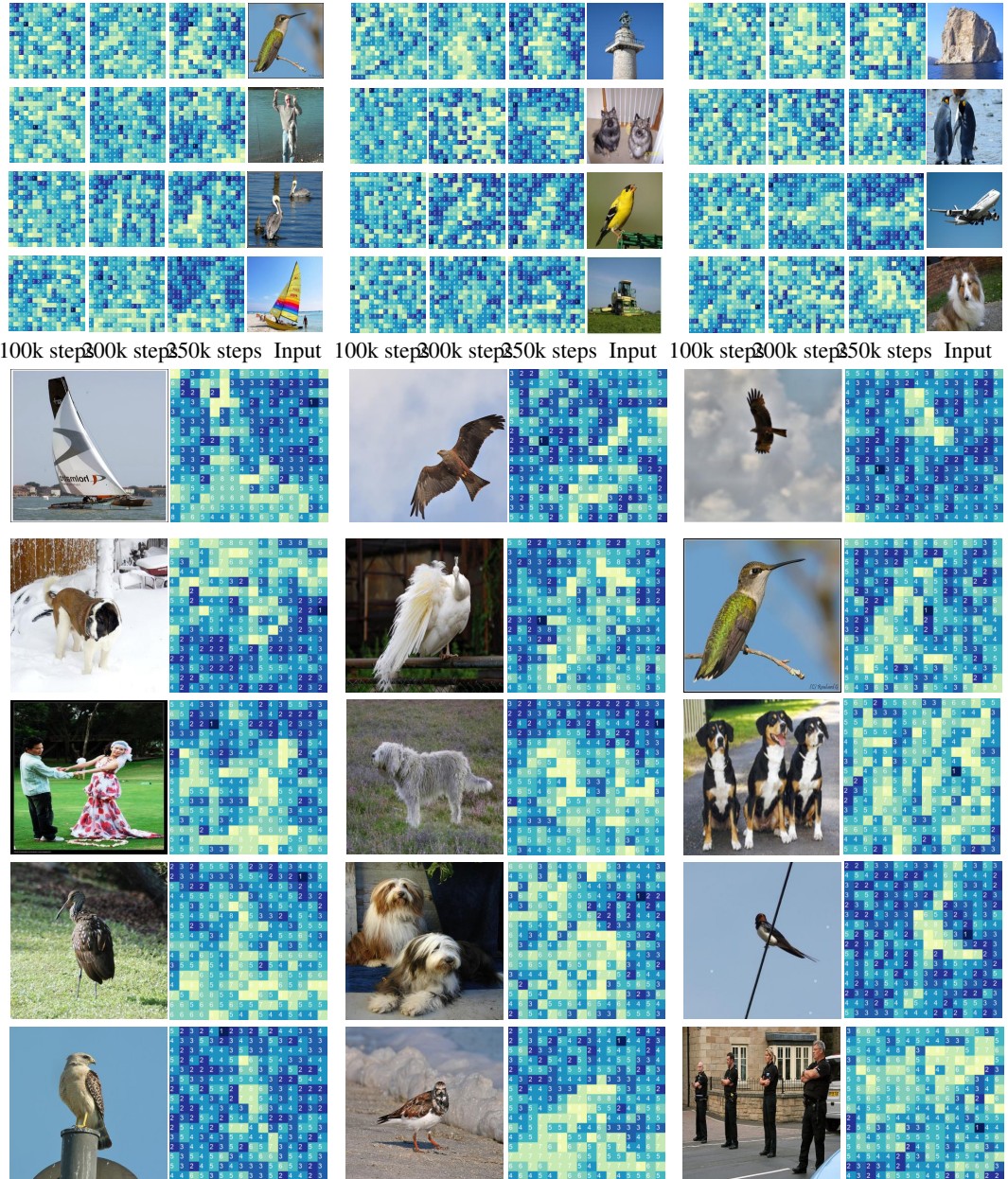

Figure 11: More visualization examples for assign diversity of sub-tokens split from different patches with respect to training steps. We analyze MH-MoE with 8 heads ($h$=8) as the subject. Brighter regions indicate that sub-tokens from this patch are distributed to a greater number of diverse experts, while darker regions indicate that sub-tokens are assigned to more of the same experts.

