# OpenReview forum: "Multi-Head Mixture-of-Experts"
_NeurIPS.cc/2024/Conference — NeurIPS 2024 poster_

### Official Review · Reviewer_Y3PR · 2024-06-16

**Soundness:** 2
**Presentation:** 3
**Contribution:** 3
**Rating:** 4
**Confidence:** 5

**Summary:**

The work presents an advanced MoE model architecture to further improve the expert utilization. Authors conduct comprehensive experiments on three pre-training tasks, English-focused language modeling, multi-lingual language modeling and masked multi-modality modeling with models at different scales (300M to 7B).

**Strengths:**

1) Important research problem. Better expert utilization has great potential to improve the MoE scalability.
2) Comprehensive experiments. From 300M to 7B. From NLP to CV.
3) Cool visualization. Figure 1 is very helpful in understanding why and how this model works.

**Weaknesses:**

I only have one but important concern: the end-to-end training and inference throughput. I understand that the theoretical computation and communication cost is not high, but since we are conducting a more complicated/fine-grained routing decision, the routing process may take longer in practice. Considering that MoE models are sometimes trained and deployed at very large scale, the intra-node all2all is sometimes required. Will the proposed approach slow down this?

I can increase my score to positive if this concern can be solved.

**Questions:**

See weakness.

---

> ### Author Rebuttal · Authors · 2024-08-06
>
> Thank you for your feedback :)
>
> ---
>
> >**Q1**: I only have one but important concern: the end-to-end training and inference throughput. I understand that the theoretical computation and communication cost is not high, but since we are conducting a more complicated/fine-grained routing decision, the routing process may take longer in practice. Considering that MoE models are sometimes trained and deployed at very large scale, the intra-node all2all is sometimes required. Will the proposed approach slow down this?
> >
> >I can increase my score to positive if this concern can be solved.
>
>
> **A1:** In fact, if traditional static routing [1] (i.e., using zero-padding or dropping tokens to force a uniform distribution of tokens among experts) were applied in MH-MoE, the situation would be different. Since MH-MoE uses sub-tokens for more complex and fine-grained routing decisions, the matrix dimensions required to record routing information expand accordingly during both training and inference. This results in more time-consuming matrix operations and higher all-to-all communication costs within nodes, thereby affecting the end-to-end throughput of deploying MH-MoE. We present a comparison of throughput between XMoE and MH-MoE using static routing (denoted as MH-MoE*) in Table A. Our results show that MH-MoE with static routing is slower than the baseline.
>
> To address this issue, we have implemented dynamic routing [2] (refer to the implementation: [databricks/megablocks (github.com)](https://github.com/databricks/megablocks)) in our MH-MoE model. Dynamic routing alleviates the constraints imposed by static routing on both software and hardware, enabling a more efficient and unrestricted routing strategy. Table A shows a comparison of the throughput taken by MH-MoE with dynamic routing (denoted as MH-MoE) and the baseline. We observe that using the more efficient dynamic routing approach results in end-to-end throughput for MH-MoE that is nearly identical to the baseline.
>
> This is the approach we have taken to address the issue of high end-to-end training and inference throughput in MH-MoE. If you have any further questions, we welcome continued discussion during the discussion phase. :)
>
> ***Table A. End-to-End Throughput on eight GeForce RTX 3090 GPUs, each with 24 GB of GPU memory.***
>
> | Models        | End-to-End Throughput (toks/s) |
> | ------------- | ------------------------------ |
> | SMoE Baseline | 32.21                          |
> | MH-MoE*       | 25.77                          |
> | MH-MoE        | 31.93                          |
>
> **Reference**
>
> [1] Lepikhin, Dmitry, et al. "Gshard: Scaling giant models with conditional computation and automatic sharding." arXiv 2020.
>
> [2] Gale, Trevor, et al. "Megablocks: Efficient sparse training with mixture-of-experts." PMLR 2023.

---

> > ### Comment · Reviewer_Y3PR · 2024-08-13
> >
> > Thank you for the explanation. I will maintain my score.
> > Megablock can alleviate the problem as the expert is small. Imagine that you are training a model with 1000 experts and 10T parameters, or even slightly smaller (maybe 1T). You finally need to do expert parallel on the slow network (inter-node communication). So the problem is still there.
> > I personally believe you should always think about a very large and sparse model when you want to push the boundary of MoE research further.

---

> ### Author Response · Authors · 2024-08-13
>
> Thank you for your feedback! Achieving a model with 1T or even more parameters is indeed a significant milestone.
>
> There are a few points we would like to clarify:
>
> 1. Increasing the number of experts does indeed increase the cost of communication. However, in our work, the MH-MoE has the same number of experts as the baseline SMoE. Therefore, when scaling to models with a large number of experts, MH-MoE faces the same communication challenges as SMoE.
>
> 2. The slower speed of static routing in MH-MoE is due to the need to create matrices of size $N(S^2)$ to store the routing status, where $S$ is the number of tokens needing routing. (Refer to [this code](https://github.com/facebookresearch/fairseq/blob/moe/fairseq/modules/moe/top2gate.py#L193): `combine1_sec` is the matrix, $s$ is the number of tokens, $e$ is the number of experts, and \(c\) is the capacity where $c = \frac{2s}{e}$, so the matrix size is $2e^2$. For MH-MoE, the number of tokens requiring routing is proportional to the number of heads, which affects the speed. In the implementation of MegaBlock, scatter and gather operations are optimized through kernel fusion (Refer to [this code](https://github.com/databricks/megablocks/blob/main/megablocks/layers/moe.py#L185)), eliminating the need to create this matrix and reducing the required memory to $N(S \times D)$, where $S$ is the number of tokens needing routing and $D$ is the dimension of the hidden state. Although the number of tokens increases to $S \times H$, where $H$ is the number of head, the token dimension decreases to $\frac{D}{H}$, so the overall memory requirement does not increase significantly. Regarding the amount of data transferred in communication, MH-MoE and SMoE are the same, both requiring $N(S \times D)$.

---

### Official Review · Reviewer_84mb · 2024-07-11

**Soundness:** 2
**Presentation:** 4
**Contribution:** 3
**Rating:** 5
**Confidence:** 5

**Summary:**

This paper proposes Multi-Head Mixture-of-Experts (MH-MoE), a simple yet effective routing strategy that splits each input token into multiple sub-tokens for expert routing. This operation significantly enhances the ratio of activated experts for each token, enabling more fine-grained assigning of tokens. Through extensive experimental results across different parameter scales (300M to 7B) and three 15 pre-training tasks, the paper validates the effectiveness of the proposed MH-MoE strategy.

**Strengths:**

1. The paper is well-written and easy to follow.
2. The proposed MH-MoE is simple yet effective. It is easy to be integrated into existing frameworks with MoE optimization. This improves the feasibility of the method.
3. The experiments cover both CV and NLP modeling scenarios and the results are consistently positive.

**Weaknesses:**

1. My major concern is about the experimental settings. Implementing MH-MoE based on the X-MoE instead of the vanilla SMoE is strange. The improvements shown by the experiments may not be generalizable to other MoE structures, which makes the proposed method less convincing.
2. Lack of experiments on scaling computations (activated experts). Though the paper conducts scaling experiments on scaling total parameters (number of experts), the computation remains static. The information conveyed by this kind of scaling is limited as the computation is also important in affecting the performance of MoE. I’m not expecting additional experiments as the time is limited during rebuttal. However, a computational scaling experiment would make the method more convincing.

**Questions:**

Can you provide the balance loss curves for your method and the baselines? I’m curious about the effect of your method on load balancing as it introduces more sub-tokens for routing.

---

> ### Author Rebuttal · Authors · 2024-08-06
>
> Thank you for your detailed feedback. We address your feedback point by point below.
>
> ---
>
> >**Q1**: My major concern is about the experimental settings. Implementing MH-MoE based on the X-MoE instead of the vanilla SMoE is strange. The improvements shown by the experiments may not be generalizable to other MoE structures, which makes the proposed method less convincing.
>
> **A1**: In fact, we conducted experiments on **two MoE structures**:
>
> 1. For English-focused language modeling, multilingual language modeling, and masked multimodality modeling tasks, **we implemented our MH-MoE on XMoE [1]** and conducted extensive experiments across these three tasks (ranging from 300M to 7B parameters, with 3 upstream tasks and 26 downstream tasks) to validate the effectiveness of MH-MoE.
> 2. For pure vision tasks, as detailed in Section 5 of the main text, **we implemented MH-MoE on AdaMV-MoE [2]**. We validated that MH-MoE can further enhance AdaMV-MoE's performance in vision tasks such as classification, object detection, and instance segmentation.
>
> **We chose XMoE and AdaMV-MoE because they have been proven superior to vanilla SMoE in extensive experiments [1, 2]**. **We believe that the extensive experiments on these multiple upstream & downstream tasks with the two MoE structures demonstrate the effectiveness of MH-MoE.** If you have any MoE structures of interest, please let us know during the discussion stage, and we would be happy to conduct further experiments :)
>
> **Reference**
>
> [1] Chi, Zewen, et al. "On the representation collapse of sparse mixture of experts." NeurIPS 2022.
>
> [2] Chen, Tianlong, et al. "Adamv-moe: Adaptive multi-task vision mixture-of-experts." ICCV 2023.
>
>
>
> ---
>
> >**Q2**: Lack of experiments on scaling computations (activated experts). Though the paper conducts scaling experiments on scaling total parameters (number of experts), the computation remains static. The information conveyed by this kind of scaling is limited as the computation is also important in affecting the performance of MoE. I’m not expecting additional experiments as the time is limited during rebuttal. However, a computational scaling experiment would make the method more convincing.
>
> **A2**: This is a meaningful suggestion. To explore the issue you mentioned, we conducted additional experiments with top-k=3 and top-k=4 to investigate the effects of increasing activated experts on MH-MoE.
>
> We performed a language model experiment based on the RedPajama dataset, utilizing a 6-layer transformer architecture with each layer having a hidden dimension of 768. For the MoE settings, the model selects the top-k=3 or top-k=4 experts for each input. We employ a Multi-Head Mixture of Experts (MH-MoE) configuration with 2 heads. The learning rate is set to 6e-4, token number of each batch is 0.5M. **Due to time and resource constraints, we present results for the model at 25k steps.**
>
> The experimental results are shown in the Table A below. We find that, **regardless of whether k=2 （see in the maintext）, k=3, or k=4, MH-MoE consistently outperforms the baseline MoE model.**
>
> ***Table A.***
>
> | Models        | k=3       | k=4  |
> | ------------- | --------- | ---- |
> | XMoE Baseline | 13.22     |   13.14   |
> | MH-MoE        | **12.68** |   **12.56**   |
>
> ---
> >**Q3**:  Can you provide the balance loss curves for your method and the baselines? I’m curious about the effect of your method on load balancing as it introduces more sub-tokens for routing.
>
> **A3**: **In the PDF file located in the Author Rebuttal at the top of the page**, we provide a comparison of the changes in balance loss during training for XMoE Baseline and MH-MoE under two training settings, 32 experts and 8 experts, in the English-focused language modeling experiment. If you have any questions, please feel free to reach out to us during the discussion phase :).

---

> > ### Comment · Reviewer_84mb · 2024-08-14
> >
> > Thank the authors for providing more results, I will keep my relatively positive rating.

---

> ### Author Response · Authors · 2024-08-14
> **Kindly Reminder by Submission1607 Authors**
>
> Dear Reviewer 84mb,
>
> Thank you again for reviewing our work. As the discussion period ends shortly, we wanted to check if you have any further questions or found our responses helpful? We are more than willing to extend our conversation and eagerly anticipate any further discussions that may arise. Please let us know and thanks for your time :).
>
> Best regards,
>
> All Authors

---

### Official Review · Reviewer_Y9UT · 2024-07-12

**Soundness:** 2
**Presentation:** 3
**Contribution:** 2
**Rating:** 4
**Confidence:** 4

**Summary:**

This paper introduces Multi-Head Mixture-of-Experts (MH-MoE), a method for training MoE models with enhanced expert activation. In particular, MH-MoE splits each input token into multiple sub-tokens, which are processed by a set of experts in parallel, and then merges them back into their original token form. The results show improved performance over dense and X-MoE models in parameter-matching settings.

**Strengths:**

The authors propose a simple yet effective idea of splitting tokens into sub-tokens to boost expert activation. The paper is clear and well-written. The authors verify their approach on a variety of tasks, including English-focused language modeling, multi-lingual language modeling, and masked multi-modality modeling, across different parameter scales (ranging from 300M to 7B by scaling the number of experts).

**Weaknesses:**

1. Experiments are performed on a limited number of models and baselines, making the results less convincing.

- The experiments are limited by a single model architecture. The authors apply their approach by modifying the previously released X-MoE model.

- The authors compare their performance with the mentioned X-MoE model, whereas there are a variety of MoE approaches that have been proposed and released recently.

- The ablation studies are limited. The performance of SMoE heavily depends on the choice of hyper-parameters, one of the most important being the number of experts to be activated (referred to as top-k). How does the activation map change with an increase in k? Would MH-MoE still perform better, or is there a saturation point? In other words, does MH-MoE seem superior to X-MoE only because a small k=2 was chosen?

2. There is a lack of discussion on limitations. The authors claim that their approach does not have any limitations. I would urge the authors to reflect on the guidelines and reconsider their answer. For example, the weaknesses mentioned above (but not limited to) should be considered.

**Questions:**

Please see the reasons to reject.

**Limitations:**

No. authors did not disclose any limitations of their method. See "Weaknesses" section for more details.

---

> ### Author Rebuttal · Authors · 2024-08-06
>
> Thank you for your feedback. We address your feedback point by point below.
>
> ---
>
> >**Q1**: The experiments are limited by a single model architecture. The authors apply their approach by modifying the previously released X-MoE model. The authors compare their performance with the mentioned X-MoE model, whereas there are a variety of MoE approaches that have been proposed and released recently.
>
> **A1**: In fact, we conducted experiments on **two MoE structures**:
>
> 1. For English-focused language modeling, multilingual language modeling, and masked multimodality modeling tasks, **we implemented our MH-MoE on XMoE [1]** and conducted extensive experiments across these three tasks (ranging **from 300M to 7B** parameters, with **3 upstream tasks and 26 downstream tasks**) to validate the effectiveness of MH-MoE.
> 2. For pure vision tasks, as detailed in Section 5 of the main text, **we implemented MH-MoE on AdaMV-MoE [2]**. We validated that MH-MoE can further enhance AdaMV-MoE's performance in vision tasks such as **classification, object detection, and instance segmentation**.
>
> We chose XMoE and AdaMV-MoE because they have been proven superior to vanilla SMoE in extensive experiments [1, 2]. **We believe that the extensive experiments on these multiple upstream & downstream tasks with the two MoE structures demonstrate the effectiveness of MH-MoE.** If you have any MoE structures of interest, please let us know during the discussion stage, and we would be happy to conduct further experiments :)
>
> **Reference**
>
> [1] Chi, Zewen, et al. "On the representation collapse of sparse mixture of experts." NeurIPS 2022.
>
> [2] Chen, Tianlong, et al. "Adamv-moe: Adaptive multi-task vision mixture-of-experts." ICCV 2023.
>
> ---
> >**Q2**: The ablation studies are limited. The performance of SMoE heavily depends on the choice of hyper-parameters, one of the most important being the number of experts to be activated (referred to as top-k). How does the activation map change with an increase in k? Would MH-MoE still perform better, or is there a saturation point? In other words, does MH-MoE seem superior to X-MoE only because a small k=2 was chosen?
>
> **A2**: This is a meaningful suggestion. To explore the issue you mentioned, we conducted additional experiments with k=3 and k=4 to investigate the effects of increasing k on MH-MoE.
>
> We performed a language model experiment based on the RedPajama dataset, utilizing a 6-layer transformer architecture with each layer having a hidden dimension of 768. For the MoE settings, the model selects the k=3  or k=4 experts for each input. We employ a Multi-Head Mixture of Experts (MH-MoE) configuration with 2 heads. The learning rate is set to 6e-4, token number of each batch is 0.5M. **Due to time and resource constraints, we present results for the model at 25k steps.**
>
> The experimental results are shown in the Table A below. We find that, **regardless of whether k=2 （see in the maintext）, k=3, or k=4, MH-MoE consistently outperforms the baseline MoE model. Additionally, there is no indication of a saturation point in MH-MoE’s performance as k increases.**
>
> The activation map's behavior with an increase in k shows effects similar to those observed in Figure 6 of the main text. Specifically, as k increases, the activation map becomes brighter, indicating that experts are activated more frequently. This is because an increase in k means that each expert's probability of being assigned tokens also increases with each token assignment. Consequently, within a fixed number of training steps, the number of times experts are activated naturally increases.
>
> ***Table A.***
>
> | Models        | k=3       | k=4  |
> | ------------- | --------- | ---- |
> | XMoE Baseline | 13.22     |   13.14   |
> | MH-MoE        | **12.68** |   **12.56**   |
>
> ---
>
> > **Q3**: There is a lack of discussion on limitations. The authors claim that their approach does not have any limitations. I would urge the authors to reflect on the guidelines and reconsider their answer. For example, the weaknesses mentioned above (but not limited to) should be considered.
>
> **A3**: Thank you for your suggestion. We will include an explicit limitation section in the latest version, summarized as follows:
>
> An interesting observation is that the performance of our MH-MoE does not consistently improve with an increasing number of heads. We hypothesize that this may be due to each sub-token's dimension becoming too small when the number of heads is too large, leading to information loss within the tokens. This, in turn, makes meaningful and effective routing more challenging, thereby affecting the overall performance of the model. This is an intriguing area for further exploration, and we plan to investigate it in future research.

---

> > ### Comment · Reviewer_Y9UT · 2024-08-13
> > **Thank you for your response**
> >
> > -> A1:
> > Some recent SOTA MoE models include SMEAR [1], BTX [2], DeepSeekMoE [3], DEMix [4], etc
> > There were two points raise in the question: 1) building on top of other models; 2) baselining against SOTA models.
> > I understand that it is impossible to compare performance with all the variety of MoE models that where developed over the last two years (since 2022 for XMoE and 2023 for AdaMV), but it is always beneficial to select those most recent models of similar architectures that 1) were evaluated on the same datasets used in the paper to understand where SOTA is; 2) architectures that aimed to solve same issue (activation sparsity).
> > Comparison against two selected baselines creates false premise that activation problem has not been challenged, and doesn't give enough information about where the proposed method lies against SOTA results (even though it is not necessary that the proposed method must improve them).
> >
> > 1. Soft Merging of Experts with Adaptive Routing
> > 2. Branch-Train-MiX: Mixing Expert LLMs into a Mixture-of-Experts LLM
> > 3. Deepseekmoe: Towards ultimate expert specialization in mixture-of-experts language models
> > 4. DEMix Layers: Disentangling Domains for Modular Language Modeling
> >
> > -> A2
> > I understand that in the limited time of rebuttal, it is impossible to perform convincing ablation studies to answer the raised questions. Initial results shared are promising, and I encourage authors to continue investigations.

---

> > > ### Author Response · Authors · 2024-08-14
> > > **Response to Reviewer Y9UT**
> > >
> > > Dear  Reviewer Y9UT,
> > >
> > > Thank you for your response and reviewing our work.
> > >
> > > ---
> > >
> > > >Q1: Baselining against SOTA models (SMEAR [1], BTX [2], DeepSeekMoE [3], DEMix [4], etc).
> > >
> > > We understand that comparing our work against SOTA MoE models to better understand where SOTA currently stands could indeed enrich our paper. However, the experiments conducted on MH-MoE across models ranging from 300M to 7B parameters were aimed at demonstrating that our proposed MH-MoE can improve the performance of MoE models at various parameter scales, especially when there are many experts, as shown in Figure 7. **This does not imply that our experimental results can be directly compared with the current SOTA models of similar scale (e.g., 7B)**, as these SOTA MoE models differ significantly from our model in terms of training data, training steps, and **especially the scale of activated experts** (e.g., some SOTA models activate 2B parameters, whereas our MH-MoE activates only a few hundred million). Therefore, introducing these results for comparison may not provide meaningful insights.
> > >
> > > Furthermore, we want to reiterate that our experiments were conducted under two different MoE frameworks, **with a total of four pretraining tasks and 29 downstream tasks** to validate the effectiveness of MH-MoE. This experimental setup is more extensive than some previous works [1-3], and we believe that our results sufficiently demonstrate the effectiveness of MH-MoE.
> > >
> > > **Reference**
> > >
> > > [1] Chi, Zewen, et al. "On the representation collapse of sparse mixture of experts." NeurIPS 2022.
> > >
> > > [2] Chen, Tianlong, et al. "Adamv-moe: Adaptive multi-task vision mixture-of-experts." ICCV 2023.
> > >
> > > [3] Soft Merging of Experts with Adaptive Routing
> > >
> > > ---
> > >
> > > >Q2: Compare with architectures that aimed to solve the same issue.
> > >
> > > Thank you for your suggestion. As far as we know, we are the first to identify the issue of low expert activation ratio in MoE models, so we are currently unable to introduce comparisons with similar models in the paper. However, we look forward to more research focusing on this issue in the future, as it is a critical challenge, especially when aiming to increase the number of experts in MoE models significantly. Low expert activation could become a bottleneck limiting the further improvement of MoE models, and we would be very interested in comparing our work with future studies addressing this issue.
> > >
> > > ---
> > >
> > > >Q3: Initial results shared are promising, and I encourage authors to continue investigations.
> > >
> > > Thank you for your positive feedback. We will continue our experiments and update the latest results in the upcoming version of the paper.
> > >
> > > ---
> > > **If you have any further questions or concerns, please feel free to contact us at any time. We are always available and look forward to further discussions with you. :)**
> > >
> > > Best regards,
> > >
> > > All Authors

---

### Official Review · Reviewer_Rrmq · 2024-07-13

**Soundness:** 3
**Presentation:** 2
**Contribution:** 2
**Rating:** 7
**Confidence:** 4

**Summary:**

The paper proposed a new architectural extension to large pre-trained models (e.g. LLMs, or Multi-modal models). Specifically, the focus is on Mixture of Experts (MoE) models and the paper proposes a new method by introducing subtokens and routing each subtoken to experts.

The resulting architectural change performs better on down-stream tasks and demonstrates a higher use of a variety of experts.

The author's also conduct a variety of ablation studies to test various aspects and changes of their proposed architectural change.

**Strengths:**

Strengths:
- Achieves strong performance on down-stream tasks and seems to out-perform the baseline methods presented in the paper.
- Paper claims that they achieve better results with same cost
- Good reporting on experiments for reproducibility
- Decent amount of experiments and evaluations

**Weaknesses:**

Weaknesses:
1. No actual comparison in terms of "cost" is given except for parameter count and complexity calculations in the appendix. It would be interesting to see actual evaluation of cost (e.g. computational cost of training & computational cost of inference vs. MoE alone).
2. Evaluation and comparison is not done on publicly available and popular (& stat-of-the-art) models of relevant sizes (e.g. Mistral, Aya, Gemma ...). Therefore, it is not clear if this method actually performs better or just on the models and training runs presented in this paper.
3. Clarity
a. some parts are unclear, e.g. equations 2,3,6,7 speak about the new additions of the method, however, it is not clear at all that there is MLP layer present (only later in the paper it is mentioned), however, it is "still" not clear whether and how it applied during equations 2,3,6,7.
b. some mistakes in the paper e.g. Paper checklist 7, the answer is N/A. However, N/A means no experiments were run, the correct answer should be No.)
4. The core claim that the problem with MoE is: "low experts activation issue" has a few problems:
a. It is never clearly explained why this is an issue (and no evidence is given that this is a real issue).
b. On one hand the paper presents that the method increases activations, however, the paper also shows that increasing the number of heads (and therefore activations) seems to also decrease performance after a specific number of heads [Figure 5, Section 4.4]. (Great for reporting this, however, this also challenges the introduction / proposition of the method). I.e. it seems that "low experts activation" is not always a problem, or only a problem to a certain degree. => Therefore, it would be interesting to revise this hypothesis, and also to provide a counter hypothesis (with evidence).
5. Reporting on sharding of the models on the DGX-2 system is not quite described. (Which is quite essential to run it sensibly.)

**Questions:**

Questions:
1. How does your model compare against some popular & state-of-the-art models (e.g. MoE LLMs like Mistral) in terms of performance?
2. Referring to "Weakness 4.b", what is your explanation of why your method actually works better?
3. What is the actual cost of running standard MoE vs. yours (in terms of practical training or inference time) - given the same computation budget.

**Limitations:**

Limitations:
1. Limitations are not really considered in the paper.
2. It would be really interesting to understand what does NOT work in the method in practice?
a.  What did not work during developing of the method?
b. When is the method worse than previous other methods?
c. Computational limitations?
d. etc. etc.

---

> ### Author Rebuttal · Authors · 2024-08-06
>
> We are grateful to the reviewer for the extensive review. Due to space constraints, we address your questions in the **Rebuttal** below as well as at the top of this page in the **Author Rebuttal**:
>
> ---
> >**Q1**: What is the actual practical training or inference time of running MoE vs. MH-MoE.
>
> **A1**: **For training cost evaluation**: In fact, if traditional static routing [1] (i.e., using zero-padding or dropping tokens to force a uniform distribution of tokens among experts) were applied in MH-MoE, the situation would be different. Since MH-MoE uses sub-tokens for more complex and fine-grained routing decisions, the matrix dimensions required to record routing information expand accordingly during training. This results in more time-consuming matrix operations and higher all-to-all communication costs within nodes, thereby affecting the end-to-end throughput of training MH-MoE. We present a comparison of throughput between XMoE and MH-MoE using static routing (denoted as MH-MoE*) in Table A. Our results show that MH-MoE with static routing is slower than the baseline.
>
> To address this issue, we have implemented dynamic routing [2] (refer to the implementation: [databricks/megablocks (github.com)](https://github.com/databricks/megablocks)) in our MH-MoE model. Dynamic routing alleviates the constraints imposed by static routing on both software and hardware, enabling a more efficient and unrestricted routing strategy. Table A shows a comparison of the training throughput taken by MH-MoE with dynamic routing (denoted as MH-MoE) and the baseline. We observe that using the more efficient dynamic routing approach results in end-to-end training throughput for MH-MoE that is nearly identical to the baseline.
>
> ***Table A. End-to-End Throughput on eight GeForce RTX 3090 GPUs.***
>
> |Models|End-to-EndThroughput(toks/s)|
> |--|--|
> |SMoEBaseline|32.21|
> |MH-MoE*|25.77|
> |MH-MoE|31.93|
>
> **For inference cost evaluation**, we implemented our MHMoE model on the Mistral 7*8B architecture and used vLLM (https://github.com/vllm-project/vllm) to test the inference speed. vLLM is a widely used LLM inference framework. We tested the inference speed on four A6000 GPUs. All GPUs processed the same prompts and used the same sampling parameters (temperature=0.8, top_p=0.95), with the tensor parallel size set to 4. To ensure the fairness of the inference speed test, given that the MHMoE model outputs garbled text, we set a consistent output length. The results are shown in Table B. We can observe that the inference speed of the MHMoE model is slightly slower than that of the SMoE model. Additionally, as the number of heads in MHMoE increases, the inference speed decreases slightly. However, the difference is not significant, and the MHMoE model still achieves a high inference speed.
>
> ***Table B.***
>
> ||InputSpeed(toks/s)|OutputSpeed(toks/s)|
> |------|--------|--------|
> |SMoE|18.47|45.47|
> |MHMoE(head=2)|18.25|44.92|
> |MHMoE(head=4)|18.18|44.74|
>
> **Reference**
>
> [1] Lepikhin, Dmitry, et al. "Gshard: Scaling giant models with conditional computation and automatic sharding." arXiv 2020.
>
> [2] Gale, Trevor, et al. "Megablocks: Efficient sparse training with mixture-of-experts." PMLR 2023.
>
> ---
> > **Q2**: Evaluation and comparison is not done on publicly available models of relevant sizes (e.g. Mistral).
>
> **A2**: **The training data and steps for the models you mentioned are either lengthy or not well-defined, making them difficult to compare directly**. To ensure fairness, we have used our own implementation of the **two MoE methods** for comparison:
>
> 1. For English-focused language modeling, multilingual language modeling, and masked multimodality modeling tasks, **we implemented our MH-MoE on XMoE [1]** and conducted extensive experiments across these three tasks (ranging from 300M to 7B parameters, with 3 upstream tasks and 26 downstream tasks) to validate the effectiveness of MH-MoE.
> 2. For pure vision tasks, as detailed in Section 5 of the main text, **we implemented MH-MoE on AdaMV-MoE [2]**. We validated that MH-MoE can further enhance AdaMV-MoE's performance in vision tasks such as classification, object detection, and instance segmentation.
>
> We chose XMoE and AdaMV-MoE because they have been proven superior to vanilla SMoE in extensive experiments. **We believe that the extensive experiments on these multiple upstream & downstream tasks with the two MoE structures demonstrate the effectiveness of MH-MoE.**
>
> ---
>
> >**Q3**: Clarity a. some parts are unclear.
>
> **A3**: Thank you for pointing this out. We apologize for the unclear or incorrect descriptions in a and b. In fact, in all experimental setups (except for the ablation experiments on MLP layers in Table 5 and the number of MLP layers in Table 13), the MH-MoE model uses both the multi-head layer and the merge layer. These two MLP layers are represented by the $W_\{head\}$ and $W_{merge}$ matrices in Equations 2 and 7, respectively. The multi-head layer and the merge layer perform matrix transformations on the last dimension (hidden state dimension) of the input tokens and the merged output tokens for feature modeling. We will revise these sections in the updated version of the paper to ensure a clearer and more accurate presentation.
>
> ---
> >**Q4**: The core claim that the problem with MoE is: "low experts activation issue" has a few problems: a. It is never clearly explained why this is an issue. b. the paper also shows that increasing the number of heads (and therefore activations) seems to also decrease performance after a specific number of heads [Figure 5, Section 4.4]. (Great for reporting this, however, this also challenges the introduction / proposition of the method). I.e. it seems that "low experts activation" is not always a problem, or only a problem to a certain degree?
>
> **A4**: Pleasee see in the **Author Rebuttal section**.
>
> ---
> >**Q5**: Reporting on sharding of the models.
>
> **A5**:  Pleasee see in the **Author Rebuttal section**

---

> > ### Comment · Reviewer_Rrmq · 2024-08-08
> >
> > Dear Author(s),
> >
> > Thank you for the detailed rebuttal and response.
> >
> > > A1:
> >
> > Comment: Thank you for providing such clear and detailed response to this question.
> >
> > Questions:
> > a) Would dynamic routing improve the Tok/s for the baseline?
> >
> > > A2:
> >
> > Comment: thank you for such a clear answer. This is very clear now.
> >
> > > A3:
> >
> > Comment: Thank you for clarifying this point. Yes, please add a revised version to make this clearer.
> >
> > > A4:
> >
> > Comment:
> > a) Thank you for clarifying why low activation of experts might be an issue. This indeed makes it clearer - perhaps the paper could reflect this answer and make this point a bit clearer.
> > b) Interesting observation. Thanks for commenting this as well.
> >
> > Questions:
> > a). While low-activation of experts seems quite plausible to be the cause of low performance for SMoE, it is not entirely proven. It would be interesting to add some simple baseline experiment that would somehow "force" the model to activate more experts and therefore increase performance - to really demonstrate that this is the matter. Alternatively (or rather additionally) it would be interesting to add additional hypothesis. (E.g. perhaps learning dynamics become very unstable, etc. - as why would SMoE perform worse with more experts, if it could just have a few "dead" experts and then it would be equivalent to an SMoE with less experts).
> > b). Similarly, with regards to number of heads and why performance starts dropping. This is an excellent observation, some validation of this hypothesis might be interesting.
> >
> > > A5:
> >
> > Comment: Thank you for providing initial information. Indeed this would be interesting to see in practice. Are there any references for your particular implementation?
> >
> > Thank you again.

---

> > > ### Author Response · Authors · 2024-08-12
> > > **Response to Reviewer Rrmq**
> > >
> > > Thanks for your detailed feedback!
> > >
> > > ----
> > >
> > > >**Q1**: Would dynamic routing improve the Tok/s for the baseline?
> > >
> > > **A1**: In Table A, we compare the speed of the baseline and MH-MoE using static routing and dynamic routing (marked with #). We observed the following:
> > >
> > > 1. Dynamic routing also increases the tokens per second (Tok/s) for the baseline.
> > > 2. The improvement dynamic routing brings to MH-MoE is significantly greater (6.16 toks/s) than the improvement it brings to the baseline (1.77 toks/s).
> > > 3. Although dynamic routing increases the Tok/s for the baseline, the improved baseline speed is not much faster than MH-MoE#.
> > >
> > > The above results indicate that dynamic routing is an effective way to address the throughput limitations of MH-MoE.
> > >
> > > ***Table A. End-to-End Throughput on eight GeForce RTX 3090 GPUs, each with 24 GB of GPU memory.***
> > >
> > > | Models         | End-to-End Throughput (toks/s) |
> > > | -------------- | ------------------------------ |
> > > | SMoE Baseline  | 32.21                          |
> > > | SMoE Baseline# | 33.98                          |
> > > | MH-MoE         | 25.77                          |
> > > | MH-MoE#        | 31.93                          |
> > >
> > > ---
> > >
> > > >**Q2**: Thank you for clarifying why low activation of experts might be an issue. This indeed makes it clearer - perhaps the paper could reflect this answer and make this point a bit clearer.
> > >
> > > **A2**: Certainly, we will clarify this point in the latest version of our paper. Thank you very much for your suggestion.
> > >
> > >
> > >
> > > ---
> > >
> > > > **Q3**: It would be interesting to add some simple baseline experiment that would somehow "force" the model to activate more experts and therefore increase performance - to really demonstrate that this is the matter.
> > >
> > > **A3**: We designed a small experiment using an MoE model based on the SMoE baseline with a 3-layer transformer, trained for 20k steps. The experiment includes two groups:
> > >
> > > 1. An MoE with 128 experts and top-1 activation (denoted as SMoE-128-top1).
> > > 2. An MoE with 256 experts and top-2 activation (denoted as SMoE-256-top2).
> > >
> > > The corresponding perplexity results are shown in Table B. We observed that SMoE-256-top2, which activates more experts (two experts per token), performs better. This demonstrates that activating more experts can provide greater benefits to the model. We will include this small experiment and its explanation in the revised version of the paper to clearly demonstrate that activating more experts makes a significant difference.
> > >
> > > ***Table B.***
> > >
> > > |               | PPL$\downarrow$ |
> > > | ------------- | --------------- |
> > > | SMoE-128-top1 | 16.31           |
> > > | SMoE-256-top2 | **15.33**       |
> > >
> > >
> > >
> > > ---
> > >
> > > > **Q4**: Alternatively (or rather additionally) it would be interesting to add additional hypothesis. (E.g. perhaps learning dynamics become very unstable, etc. - as why would SMoE perform worse with more experts, if it could just have a few "dead" experts and then it would be equivalent to an SMoE with less experts)
> > >
> > > **A4**: Thank you for your interesting suggestion. One hypothesis we have is that when there are more experts, insufficient training of the experts may lead to underfitting, resulting in reduced model performance. We will continue to explore possible underlying reasons and include these hypotheses and explanations in the revised version of the paper to enrich the discussion.
> > >
> > > ---
> > >
> > > > **Q5**: Similarly, with regards to number of heads and why performance starts dropping. This is an excellent observation, some validation of this hypothesis might be interesting.
> > >
> > > **A5**: This is indeed a very interesting phenomenon. We also noted that previous studies have reported similar observations in multi-head self-attention, where increasing the number of heads does not necessarily lead to better performance. We are conducting further experiments to explore the essence of this phenomenon and the potential insights it might offer.
> > >
> > > We can offer a hypothesis here: when the number of heads increases, the dimension of each head decreases, which may harm the representation power of each head. This can cause each head to suffer from a **low-rank bottleneck**, leading to reduced model performance. The impact of this low-rank bottleneck has been studied in the context of multi-head self-attention [1], and we are conducting similar research on MH-MoE in the future to further explore the reasons behind this phenomenon.
> > >
> > > **Reference**
> > >
> > > [1] Bhojanapalli, Srinadh, et al. "Low-rank bottleneck in multi-head attention models." PMLR, 2020.
> > >
> > > ---
> > >
> > > >**Q6**: Are there any references for your particular implementation?
> > >
> > > **A6**: We referenced the following libraries to implement our model:
> > >
> > > [1] https://github.com/databricks/megablocks
> > >
> > > [2] https://github.com/facebookresearch/fairseq
> > >
> > > [3] https://github.com/microsoft/torchscale
> > >
> > > ---
> > > Your suggestions have provided us with a lot of new inspiration and insights. If you have any further questions, please feel free to let us know. We would be happy to discuss them further with you! :)

---

> > > > ### Comment · Reviewer_Rrmq · 2024-08-13
> > > >
> > > > Dear Authors thank you for taking the time and effort to do those additional experiments.
> > > >
> > > > Things are much clearer now and the score is now 7 (from 6) with confidence 4 (from 3).
> > > >
> > > > We would highly encourage to add all these changes into the paper (incl. appendix) and to make certain things clearer. As such this paper becomes a good contribution.

---

> > > > > ### Author Response · Authors · 2024-08-13
> > > > > **Thanks**
> > > > >
> > > > > Thank you very much for your positive feedback and your willingness to raise the score and confidence. We will ensure that the revised version of our paper incorporates all the suggested improvements to enhance its clarity and contribution :).

---

### Author Rebuttal · Authors · 2024-08-06

## Supplementary rebuttal for Reviewer Rrmq

---

>**Q4**: The core claim that the problem with MoE is: "low experts activation issue" has a few problems: a. It is never clearly explained why this is an issue. b. the paper also shows that increasing the number of heads (and therefore activations) seems to also decrease performance after a specific number of heads [Figure 5, Section 4.4]. (Great for reporting this, however, this also challenges the introduction / proposition of the method). I.e. it seems that "low experts activation" is not always a problem, or only a problem to a certain degree?

**A4**: We address your questions a and b point by point below:

(a). As shown in Figure 7 (a) and (b), we observe that when the number of experts increases significantly (e.g., to 128 or 256), the performance of the SMoE model declines both upstream and downstream. This suggests that with a large number of experts, the SMoE model suffers from an imbalance in expert activation, where most experts have very low activation rates and only a small subset of experts remain active. As a result, the majority of experts do not receive sufficient data for optimization, making it challenging for SMoE to leverage the benefits of a large number of experts. This issue highlights that the "low experts activation issue" is a major factor limiting the further scaling of the SMoE model. **Therefore, we believe that addressing the "low experts activation issue" is crucial, especially as we aim to improve model performance by increasing the number of experts to very large sizes, where this issue may become a significant bottleneck.**

(b). It is indeed possible for the model performance to first increase and then decrease as the number of heads grows. We hypothesize that, although the "low experts activation issue" is mitigated, when the number of heads is excessively large, each sub-token's dimensionality is divided too finely. This can lead to a loss of internal token information and make meaningful and effective routing more challenging, thereby affecting the overall performance of the model. This is a fascinating area for exploration, and we appreciate your observation. We will investigate this further in our future research.



---

>**Q5**: Reporting on sharding of the models.

**A5**:  We apologize for not adequately describing the reporting on sharding of the models in our paper. Specifically, we employed both **data parallelism** and **expert parallelism** strategies. We will include the corresponding details in the revised version of the paper.

---

### Decision · Program_Chairs · 2024-09-25

**Decision:**

Accept (poster)

**Comment:**

This paper presents a mixture-of-experts approach that routes different sub-tokens to different experts. Reviewers found the idea to be simple and effective, the results to be strong compared to existing baselines, the paper clear, and the experiments comprehensive. Concerns regarding missing experiments, unsupported claims and clarity issues have largely been addressed. Some concerns about XXL-scale experiments seem out of scope in my view, as there is plenty of value in models much smaller than 10T parameters. Other requests for additional experiments and baselines seem not essential either.